# Coulomb pre-stress and fault bends are ignored yet vital factors for earthquake triggering and hazard

Z.K. Mildon [1,2], G.P. Roberts[3], J.P. Faure Walker [2] & S. Toda[4]

Successive locations of individual large earthquakes ($M_w > 5.5$) over years to centuries can be difficult to explain with simple Coulomb stress transfer (CST) because it is common for seismicity to circumvent nearest-neighbour along-strike faults where coseismic CST is greatest. We demonstrate that Coulomb pre-stress (the cumulative CST from multiple earthquakes and interseismic loading on non-planar faults) may explain this, evidenced by study of a 667-year historical record of earthquakes in central Italy. Heterogeneity in Coulomb pre-stresses across the fault system is >±50 bars, whereas coseismic CST is <±2 bars, so the latter will rarely overwhelm the former, explaining why historical earthquakes rarely rupture nearest neighbor faults. However, earthquakes do tend to occur where the cumulative coseismic and interseismic CST is positive, although there are notable examples where earthquake propagate across negatively stressed portions of faults. Hence Coulomb pre-stress calculated for non-planar faults is an ignored yet vital factor for earthquake triggering.

[1] School of Geography, Earth and Environmental Sciences, University of Plymouth, Drake Circus, Plymouth PL4 8AA, UK. [2] Institute for Risk and Disaster Reduction, University College London, Gower Street, London WC1E 6BT, UK. [3] Department of Earth and Planetary Sciences, Birkbeck, University of London, Malet Street, London WC1E 7HX, UK. [4] International Research Institute of Disaster Science, Tohoku University, Aramaki Aza-Aoba 468-1, Aoba-ku, Sendai 980-0845, Japan. Correspondence and requests for materials should be addressed to Z.K.M. (email: zoe.mildon@plymouth.ac.uk)

Typically, earthquakes transfer static Coulomb stress onto the nearest neighboring faults during coseismic slip on the order of <±2 bars[1–3]. The Coulomb stress transfer (CST) is usually discussed in terms of whether an earthquake is likely to be triggered on receiver faults, especially if a so-called seismic gap[4] is identified on one or more receiver faults. However, the magnitude and spatial variability of Coulomb pre-stress (which we define as the static stress present on a brittle fault plane prior to rupture) across any particular fault is typically poorly known. It is commonly assumed to be zero when authors hypothesise which fault may be the next to experience an earthquake[1,5,6]. Other authors compare the CST to the tectonic loading rate to deduce the temporal magnitude of the advance or delay of earthquakes[7–9]. The zero or uniform value assumption is likely to be erroneous and hence misleading because we know that, firstly, interseismic stresses will have accumulated over centuries to millennia due to tectonic loading[10], secondly, multiple earthquakes over many centuries will have contributed coseismic CST[2], and thirdly, local bends in the fault geometry will have amplified or diminished the cumulative interseismic and coseismic CST[11–15]. These three factors suggest it is unlikely that Coulomb pre-stress is zero or spatially uniform as is commonly assumed when calculations of coseismic CST following large earthquakes are undertaken[5,15,16]. The question is therefore whether coseismic CST can overwhelm Coulomb pre-stress in all cases or not; the former is needed if earthquake sequences are to be explained solely with coseismic CST from single prior earthquakes. If coseismic CST cannot or rarely overwhelms Coulomb pre-stress, then the pre-stress must be taken into account when coseismic CST is calculated following large earthquakes and used to speculate on the location of future damaging earthquakes and associated seismic hazard[8].

To investigate the above, we define Coulomb stress and then study the central Apennines extensional system. Coulomb stress is defined by the following equation:

$$\Delta CST = \Delta\tau - \mu(\Delta\sigma + \Delta P) \tag{1}$$

where $\Delta CST$ is the Coulomb stress transfer, $\Delta\tau$ is the change in shear stress (in the direction of fault slip), $\mu$ is the coefficient of friction, $\Delta\sigma$ is the change in normal stress and $\Delta P$ is the change in pore fluid pressure[15,17]. Pore fluid pressure changes have been hypothesised to trigger localised earthquake sequences[18,19] but we lack direct measurements of the magnitude and spatial extent of this factor at seismogenic depths. Thus, in this paper we neglect pore fluid effects as we are interested to see if we can explain our observations without adding ad hoc fluid pressure changes and discuss this later in the paper.

The central Apennines extensional system has been studied for two reasons. Firstly, it has one of the longest known historical records of damaging earthquakes[20] (Fig. 1a), a pre-requisite to understand the accumulation of coseismic CST from multiple earthquakes. Secondly, the normal faults are well-exposed at the surface, which enables the geometry and slip rates to be accurately quantified to model variable fault geometry and interseismic loading from underlying shear zones[21–27] (Fig. 1c–e). Interseismic CST loading is modeled as an annual rate of loading (Fig. 1d). The magnitude of annual interseismic CST is low (−0.06 to 0.22 bars) compared to coseismic CST (on the order of <±2 bars, see ES6), but when summed over decades to centuries (or longer), it becomes an important component of the Coulomb pre-stress. The interseismic CST is dependent on the Holocene slip rate (measured from surface offsets) and the strike-variable fault geometries. We describe the role of detailed fault geometry in CST calculations, the differences between analyzing solely coseismic CST from single earthquakes versus cumulative coseismic and interseismic Coulomb stress (Coulomb pre-stress) over many centuries containing numerous earthquakes, finally discussing the implications for how future CST calculations should be conducted.

## Results

**The role of fault geometry.** It is known that fault geometry affects calculations of CST[12,14,28–30] but the influence of along fault variations in geometry are not routinely considered. Efforts have been made to quantify the sensitivity of the CST to the parameters of strike, dip, rake, coefficient of friction and Skempton's coefficient[11,31]. For normal faults, it is demonstrated that the CST is most sensitive to the varying strike of receiver faults[11]. Therefore it is expected that along-strike fault bends on receiver faults would amplify or diminish the CST when compared to adjacent regions of the fault. Examples are shown in Fig. 2 for four recent earthquakes ($M_w \geq 6.0$) in the central Apennines. In these examples, the difference in CST between planar fault models and non-planar fault models with along-strike bends is in the range of −2.7 to 2.4 bars. This is higher than the hypothesised CST triggering threshold of 0.1–0.5 bars[13,32,33] (although the existence of a triggering threshold is debated[34,35]). In addition, where fault bends reduce CST, this may generate negative stress barriers that could impede earthquake rupture propagation[36]. These barriers have been invoked to explain the pattern of seismicity in the 2016 central Italian earthquake sequence[37] (see Fig. 3b); without including bends in the models, these barriers would not be generated and the sequence would be difficult to explain with conventional planar and coseismic-only CST modeling (see ref. [38–40] for examples). Thus, modeling of fault bends provides valuable additional information compared to the conventional planar approach. In other words, if fault bends are not modeled, the coseismic CST calculated would not resolve important regions with raised or lowered stress at bends that may represent sites where subsequent ruptures associated with large earthquakes may nucleate or terminate.

**The role of coseismic versus cumulative Coulomb stress.** Here we study a set of 34 $M_w = 5.6–7.0$ earthquakes (from 1349–2016 A.D.) that occurred in the central Apennines rupturing 41 faults (some earthquakes ruptured more than one fault[10]). The coseismic CST of each historical earthquake and the cumulative (coseismic plus interseismic) CST prior to each earthquake have been calculated to investigate the importance of Coulomb pre-stress (see Supplementary Data 1 and Supplementary Movie 1 for the coseismic CST associated with historical earthquakes, and Supplementary Data 2 and Supplementary Movie 2 for the cumulative CST prior to each historical earthquake). Supplementary Movie 1 shows that when solely coseismic CST is considered, successive earthquakes jump around the fault system through time with no examples of nearest-neighbour faults rupturing (see Fig. 3a for an example from a single time-slice). Supplementary Movie 2 shows that the combined effect of coseismic and interseismic stress loading on non-planar faults produces significant Coulomb pre-stress heterogeneity through time (see also Fig. 3b for an example from a single time-slice). Note that no dynamic nor post-seismic stress changes are considered in this study, because the time between earthquakes is typically longer than timescales over which dynamic stress triggering will play a role[41], the magnitude of the earthquakes and timescales in this study are relatively small and therefore the effects of post-seismic stress will be negligible[42,43], and in any case the post-seismic contribution is included in the interseismic deformation we model. Moreover, post-seismic stress changes will alter the magnitude of the CST values (increase the positive stress lobes and reduce the negative stress lobes), but not change the geometry of the first-order stress pattern we describe[42,44]. The

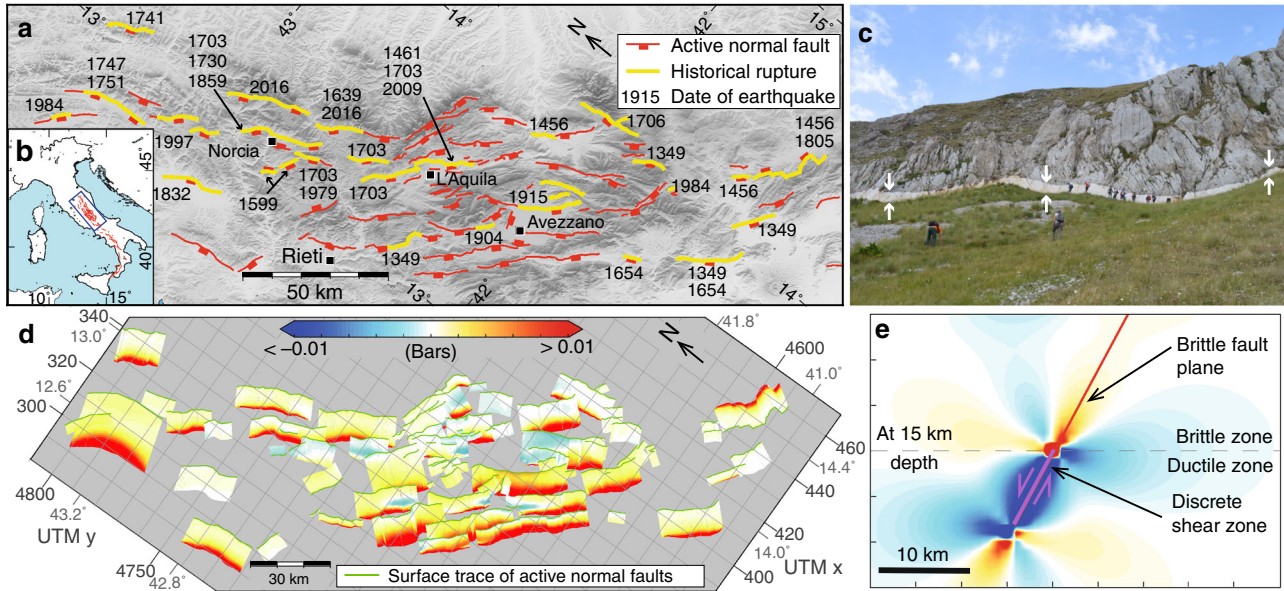

**Fig. 1** Summary of input data for this study. **a** Fault map of the central Apennines[62,63]. Towns are shown with black squares and are labeled. **b** Blue box shows the location of the study area within Italy. **c** Field photo of the coseismic rupture from two earthquakes in 2016 that both occurred on the Mt. Vettore fault. White arrows show the top and bottom of the coseismically exhumed fault scarp. **d** Heterogeneous annual interseismic loading rate on the brittle portions of non-planar faults from underlying ductile shear zones (not shown in figure). **e** Cross-section of shear zone model and associated Coulomb stress transfer (CST) for annual interseismic loading. Colour scale is the same as **d**

important point is that magnitudes of pre-stress heterogeneity (>±50 bars) are much greater than those produced solely by coseismic slip (<±2 bars). The question arises as to where subsequent earthquakes do occur given the spatial and temporal heterogeneity, and examples from the central Apennines allow us to study this.

The most recent destructive sequence of earthquakes in central Italy occurred in 2016, with three earthquakes ($M_w$ 6.1, 6.2, and 6.6)[40] along the Mt. Vettore and Laga faults (Fig. 3). The Coulomb pre-stress prior to these earthquakes is considered in two different ways: Fig. 3a shows the pre-stress solely from combined coseismic CST from three earthquakes in 1997 (Umbria-Marche seismic sequence, $M_w = 5.7$, 6.0, and 5.6) and the 2009 L'Aquila earthquake ($M_w$ 6.2); Fig. 3b shows the Coulomb pre-stress from 667 years of interseismic and coseismic CST including 31 $M_w = 5.6$–7.0 earthquakes. Both approaches produce spatial heterogeneity in transferred stress on receiver faults, but the magnitude of stress varies markedly (±1 bar compared to ±50 bars), and we note Coulomb pre-stress heterogeneity exists across the Mt. Vettore and Laga faults, which ruptured during three earthquakes in 2016 (MTV and LAG on Fig. 3).

Considering only the coseismic CST from 1997–2009 (Fig. 3a), parts of both the Mt. Vettore and Laga faults were positively stressed prior to rupture, but neither had the highest stress in the region. In addition, on these two faults the solely coseismic Coulomb pre-stress was almost entirely positive (−0.05–0.24 bars on the Mt. Vettore fault and 0.02–0.48 bars on the Laga fault) and the pattern of Coulomb stress does not appear to explain the terminations and time-sequence of the ruptures in 2016 (see the locations of the ruptures marked in Fig. 3a).

In contrast, considering the full 667-year Coulomb pre-stress, a more complicated spatial heterogeneity is generated. The Mt. Vettore fault has negative Coulomb pre-stress above 8 km on the portion of the fault that ruptured on 24 August 2016, with positive Coulomb pre-stress at depths below 8 km (negative stress of >−11.68 bars at <8 km, with 36.75 bars positive stress at

depth). The presence of shallow negative pre-stress is generated by multiple across-strike earthquakes (on the Norcia fault, e.g. in 1703[45]) and is robust across a range of slip distributions (see Supplementary Data 3). For the portion of the fault from 0–8 km depth, the most negative stress values are located within along-strike fault bends, and it has been argued that these helped to terminate ruptures in the 2016 sequence[32] (see Fig. 3b). This example is interesting because the presence of negatively stressed regions might previously have lessened concerns about rupture on this fault; however, rupture did occur within the negatively stressed region complicating the rupture sequence[40,46,47]. It may be that the change from negative to positive pre-stress at a depth of about 8 km may have been critical in allowing an earthquake to nucleate, but more examples are needed before a strong conclusion is warranted, and we investigate the locations of subsequent earthquakes later in this paper. In contrast to the Mt. Vettore fault, the portion of the Laga fault that ruptured on the 24 August was predominately positively stressed above 8 km depth (<3.52 bars) with the exception of the very northern tip of the fault that was negatively stressed (>−10.48 bars) due to the presence of a bend in the fault. Positive stress with up to 66 bars existed along the entire length of the Laga fault at depth. An important finding is therefore that during the 24 August ($M_w = 6.0$) earthquake, both positively and negatively stressed regions ruptured (based on the published slip distributions that are inverted for planar faults[46,48,49]); a similar pattern is true of the 30 October $M_w$ 6.6 earthquake (Supplementary Data 2 and Supplementary Movie 2). Thus, negatively stressed regions must not be excluded when considering the possibility of rupture.

**Analysing the historical earthquake sequence.** To further illustrate the importance of Coulomb pre-stress, in particular during fault interaction, we have examined the relationship between pre-stress on each fault and subsequent rupture for individual coseismic stress transfer events (Fig. 4a–c; Supplementary Data 1, Supplementary Movie 1 and Supplementary Data 4), and through

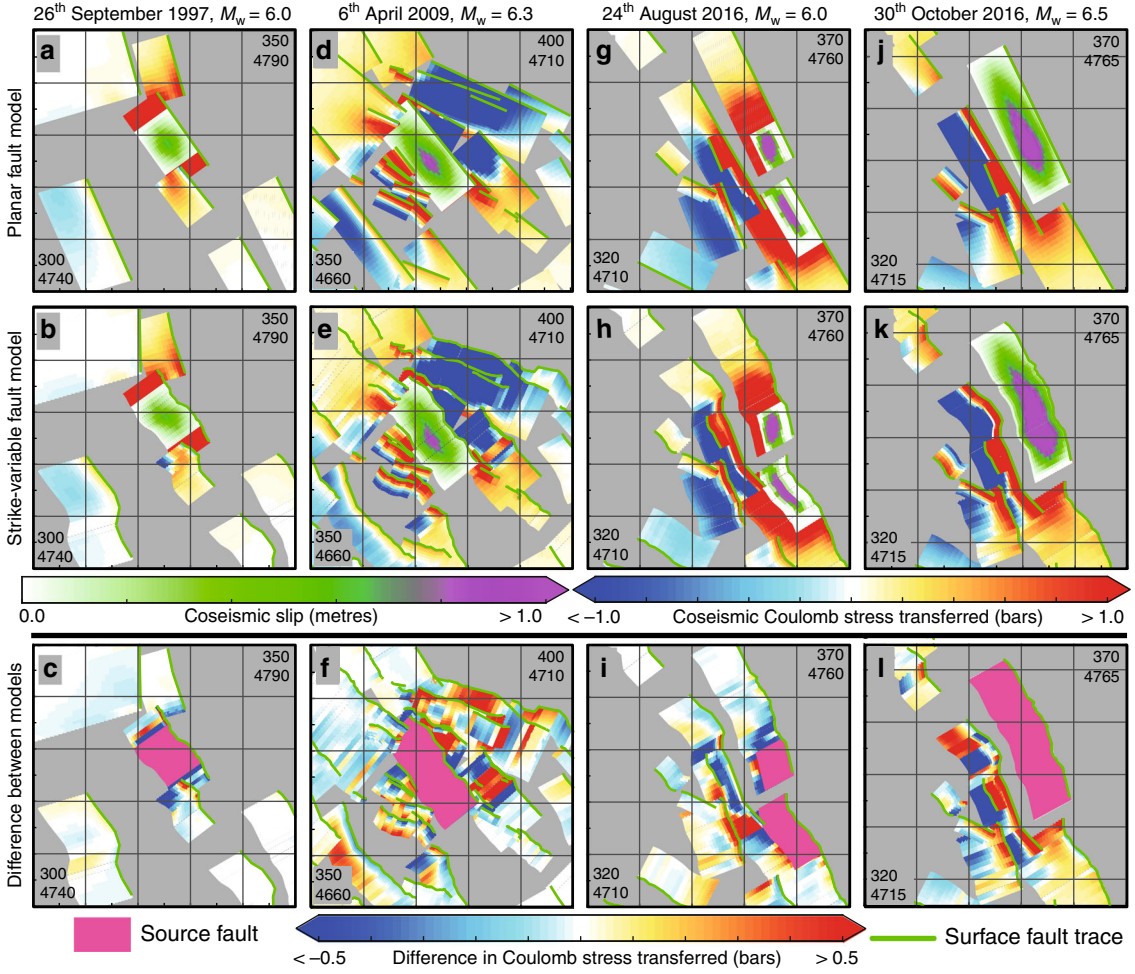

**Fig. 2** Comparing planar and strike-variable coseismic stress models. Coulomb stress transfer (CST) for four recent earthquakes; 26 September 1997 (panels **a**, **b**, **c**); 6 April 2009 (panels **d**, **e**, **f**); 24 August 2016 (panels **g**, **h**, **i**); 30 October 2016 (panels **j**, **k**, **l**). The coseismic CST and slip distributions used are shown in colour scales, UTM coordinates for the north-east and south-west corner of each panel are given (33 T zone). Panels **a**, **d**, **g**, **j** show the planar fault geometry CST models, panels **b**, **e**, **h**, **k** show the strike-variable fault geometry CST models, panels **c**, **f**, **i, l** show the difference between the coseismic CST models for the four earthquakes investigated. A positive difference indicates that including the fault geometry has increased the CST

time for the sequence of 34 earthquakes including interseismic loading since 1349 A.D. To analyse the CST, we consider the mean stress across whole fault surfaces, as well as the maximum stress on a single fault patch and the proportion of the fault that is positively stressed. We utilise these two new factors to reflect the CST heterogeneity across fault planes (which mean stress does not reflect), and use of these factors is prompted by the notion that high stress may trigger an earthquake, but the rupture needs to then be able to propagate across a fault surface[36]. Considering only coseismic CST, the proportion of faults that are positively stressed is highly variable over time (Fig. 4a, b) because this depends on the spatial arrangement of faults, with faults along-strike generally positively stressed and those across strike generally negatively stressed. In this analysis, we do not consider antithetic faults for analysis, as rupture on such faults is typically dependent on bending stresses produced by rupture of the main synthetic faults.

For individual coseismic CST, a common expectation is that following an earthquake, the next fault to rupture will be the one with the highest CST. However, we show that the next fault to rupture is never the fault with the highest mean coseismic CST (i.e. the nearest-neighbour fault, see Fig. 4b). Considering only coseismic CST for the whole historical sequence of earthquakes, 78% of the faults that rupture had mean positive coseismic CST

from the previous earthquake, but it is important to note that the magnitudes of these stresses are very small (Fig. 4b), with only three examples out of 32 having CST > 0.1 bars (a hypothesised triggering threshold[32]); this is only 9% of positive examples. Other authors argue for a triggering threshold of 0.2 bars[50], and if this threshold is considered, then only one example out of 32 (3%) had CST above this (Supplementary Data 4). Considering the maximum stress and proportion of the fault that is positively stressed (Fig. 4c), our results show that, for faults that rupture, 90% of examples have >50% of the fault plane positively stressed, but only 32% of examples had a maximum stress >0.1 bars. In addition, the mean coseismic CST on the subsequent fault that ruptures are in the range from −2.2 to 0.3 bars (Fig. 4b). These results demonstrate that solely coseismic CST does not explain the observations because the earthquake sequence misses out nearest-neighbour faults, so it is inadvisable to use solely coseismic CST to forecast the location of the next major earthquake.

Therefore, to understand the role and importance of Coulomb pre-stress, we consider the cumulative CST, comprised of coseismic CST from historical earthquakes[10,20] and interseismic CST from tectonic loading associated with underlying shear zones[10,21]. We show that summed interseismic and coseismic CST over 667 years on non-planar, strike-variable faults show

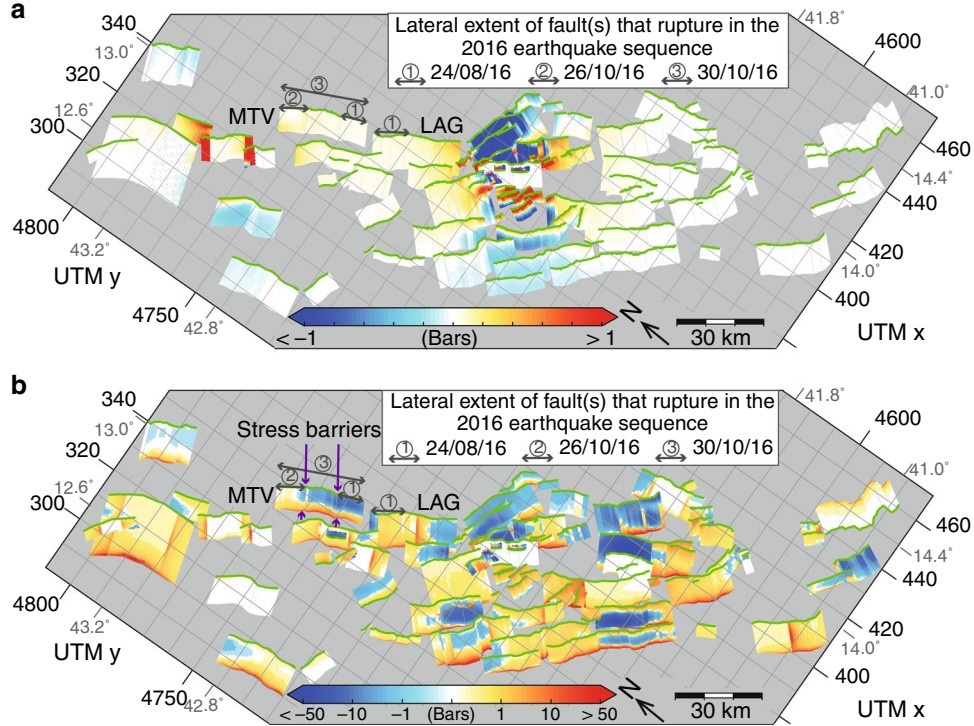

**Fig. 3** Short-term and long-term Coulomb stress models prior to 2016. **a** Coseismic CST from 1997–2009 from four earthquakes ($M_w$ = 5.7, 6.0, 5.6, and 6.3) summed together. The faults that ruptured in the sequence are labeled (MTV- Mt. Vettore, LAG- Laga) and the extent of the faults that ruptured in the three events (1. 24 August 2016, 2. 26 October 2016, 3. 30 October 2016) is shown. **b** Map of cumulative interseismic and coseismic CST from 1349 A.D. to early 2016 A.D. showing the Coulomb pre-stress prior to the 2016–17 seismic sequence. Note the non-linear colour scale of cumulative CST. The faults that ruptured in the sequence are labeled as in **a**. Purple arrows show the location of inferred negative stress barriers[37]. See Supplementary Data 1 and Supplementary Movie 1 for coseismic CST associated with historical earthquakes ($M_w$ > 5.5) since 1349 A.D. See Supplementary Data 2 and Supplementary Movie 2 for the Coulomb pre-stress on each fault prior to each historical earthquake

spatial variations of >±50 bars on individual faults (Fig. 3b, Supplementary Data 2, Supplementary Movie 2 and Supplementary Data 4), an order of magnitude larger than solely coseismic CST (Fig. 2).

To assess where earthquakes do occur within the heterogenous field of cumulative CST produced by fault and shear zone interaction (e.g. Fig. 3b), we examine earthquakes since 1703 A.D.[10]. We do not analyse earthquakes in the 354 years between 1349 A.D. and 1703 A.D., to allow the model to burn in and avoid biasing our results to the initial conditions of the model, and to allow the cumulative CST values to reflect interaction between the faults and shear zones. We can be confident of our results because the conclusions we draw are not reliant on full steady state, whose approach would be identified when all faults have had at least one earthquake, and the rate of overall release of stress in earthquakes begins to match the rate of loading. Our dataset has not achieved steady state, despite its relatively long historical record of earthquakes. This is a typical problem, and there are likely to be few, if indeed any, examples worldwide where the historical record is long enough relative to the regional deformation rate to provide data that enable study of steady state. However, we are not reliant on steady state because we simply want to study earthquake occurrence in a system where interaction is clearly underway. We are encouraged in thinking that interaction is underway by 1703 A.D. in our model for three reasons. Firstly, at least some of the faults have values of mean cumulative CST that are constant over time periods spanning several earthquakes, with across-strike CST decreases counteracting interseismic loading (Fig. 4d, e). Secondly, the percentage of faults with positive mean Coulomb pre-stress is approximately

constant after 1703 A.D., ranging between 70–76 % (Fig. 4d), supporting the hypothesis that most faults are critically stressed[51]. Thirdly, there is no obvious relationship between the percentage of positively stressed elements on a fault nor the maximum Coulomb pre-stress in relation to time, suggesting that entire fault surfaces became loaded early in the 1349–2016 A.D. time period, and before the time when we start our analysis (1703 A.D.; Fig. 4f). We are encouraged because these three points all indicate fault interaction is underway by 1703 A.D. in our model and the faults are therefore experiencing cumulative CST values that are not atypical of an interacting extensional system, so we can assess where earthquakes occur within the heterogenous field of cumulative CST produced by fault and shear zone interaction. We show below that through analysis of this sub-set of earthquake Coulomb pre-stresses, we can demonstrate that by considering the mean pre-stress and the proportion of the fault that is positively stressed, a greater proportion of the earthquake history can be understood, compared to analyzing the coseismic CST alone.

When the locations of earthquakes in the sequence of historical earthquakes from 1703–2016 A.D. are assessed, the mean cumulative CST on faults that ruptured in our sample from central Italy was positive for 23 out of 29 examples (79%), and 96% of these had a mean CST since the start of the time period we consider in the model run of >0.1 bars (the hypothesised triggering threshold[32]). Of the six examples with negative mean CST in this time period, five examples had patches (13–52% of their fault area) that were positively stressed prior to rupture (Fig. 4f). This includes the Mt. Vettore fault prior to the first earthquake in 2016 (Fig. 3b). Therefore, if both the mean cumulative CST and the existence of at least some portion of the

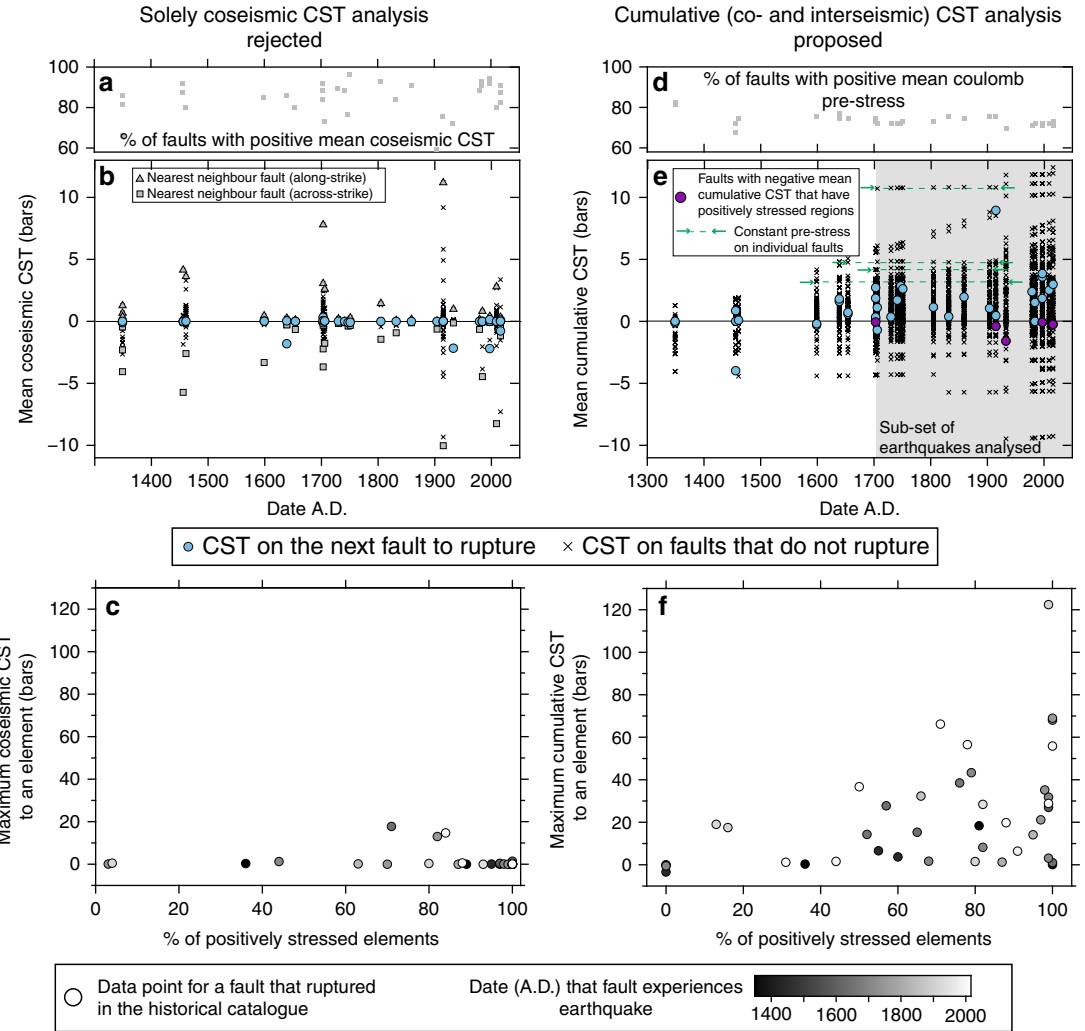

**Fig. 4** Analysing the Coulomb stress evolution on faults over time. Two approaches are considered; solely coseismic CST (rejected herein) and the proposed approach of this paper to analyse cumulative CST and Coulomb pre-stress. Vertical scales are comparable between sub-figures **a** and **d**, **b** and **c**, and **d** and **f** to assist direct comparison of the two approaches. **a** Percentage of all faults in the Apennines that have mean positive coseismic CST after each historical earthquake. **b** Mean coseismic CST on faults that do and do not rupture. The fault that ruptures (pale blue circles) never has the highest mean coseismic CST prior to rupture (grey triangles for the nearest-neighbour fault along-strike). **c** Considering the maximum stress on a single fault element and the proportion of the fault that is positively stressed prior to rupture. **d** Percentage of all faults in the Apennines that have mean positive Coulomb pre-stress prior to each earthquake in the historical catalogue. The proportion of faults that are positively stressed is fairly consistent and high, indicating that most faults are critically stressed. **e** Mean cumulative (interseismic and coseismic) CST on faults that do and do not rupture. The fault that ruptures (pale blue circles) is never the fault with the highest mean cumulative CST. The grey shaded region indicates the sub-set of earthquakes analysed (earthquakes since 1703 A.D.) to reduce the bias towards the initial conditions of the model. **f** Considering the maximum cumulative CST on a single fault element and the proportion of the fault that is positively stressed prior to rupture. The dots representing faults that rupture are colour-coded according to date, indicating that there is no correlation between time of earthquake and the maximum cumulative CST. The magnitude of cumulative CST is several orders of magnitude higher than for coseismic stress changes alone and the values are more comparable to stress drops calculated for large earthquakes. The data and analysis presented in this figure are detailed in Supplementary Data 4

fault that is positively stressed prior to rupture are considered to be criteria promoting failure, the cumulative CST results can explain 28 out of 29 (97%) examples in the historical record. This is more than can be explained by coseismic CST alone, where only 9% of the positively stressed faults had >0.1 bars CST (the hypothesised triggering threshold[32]). Furthermore, it is important to re-emphasise that the magnitudes of mean and maximum cumulative CST from the 1703 A.D. to present-day sample (Fig. 4e, f) are greater than the magnitude of CST from a single earthquake (Fig. 4b, c), showing that it is unlikely that the coseismic CST from a single earthquake will be able to overcome the Coulomb pre-stress generated by historical earthquakes and interseismic loading.

## Discussion
The modelling presented in this paper takes account of static elastic Coulomb stress transfer from historical earthquakes and interseismic loading (driven by down-dip viscous shear zone extensions of the brittle faults). There are other factors that may affect triggering of earthquakes on short and long timescales.

The long-term viscoelastic response of earthquakes has been hypothesised to affect centennial-scale earthquake triggering effects in the central Apennines[43]. In particular the relationship between the 1915 Fucino earthquake (the largest magnitude earthquake to affect the region in the historical record) and the 2009 L'Aquila earthquake has been investigated; there is a spatial separation of ~35 km between these two events. The magnitude of

post-seismic stress from the 1915 earthquake on the Paganica fault (responsible for the 2009 earthquake) is calculated to be ~0.6 bar[43], approximately twice the hypothesised magnitude of inter-seismic loading from this study (~0.37 bars) over the same time period. Over this relatively short time period, these results suggest that post-seismic and interseismic stress have similar importance for tectonic loading. However, most earthquakes in central Italy are $M < 7$ and therefore the post-seismic stress changes associated with most earthquakes will be smaller. For example, the post-seismic stress changes associated with an M6 will be an order of magnitude smaller, thus an order of magnitude smaller than the corresponding interseismic loading. In addition, the magnitude of post-seismic stress changes over this relatively short time period are an order of magnitude smaller than the magnitudes of cumulative CST calculated in this paper over centuries. Therefore, we surmise that the effects of post-seismic stress are small compared to the magnitude of Coulomb pre-stress. Also, we have neglected fluid pressure changes in this study (see Equation 1) and yet managed to explain that 97% of historical earthquakes occur on faults that have positive mean cumulative CST or at least some portion of the fault that is positively stressed. This suggests that fluid pressure changes are not required to produce the effects we have described, although we have not ruled out some fluid pressure influences, and this requires more study.

This study demonstrates that the studied active faults are interacting via the mechanism of static Coulomb stress transfer, which accumulates over multiple coseismic events and through interseismic loading. These values imply that insights into whether future earthquakes will be triggered by past earthquakes are unlikely to be gained solely from studies of coseismic CST from single earthquakes. It is especially important to note that in the sequence of events studied, the next fault that ruptures is never the nearest-neighbour fault; this is often assumed in seismic hazard assessment when discussing the likelihood of triggering[8]. Without knowledge of Coulomb pre-stress and its spatial heterogeneity, one cannot conclude that coseismic CST will over-whelm Coulomb pre-stress. We emphasise that Coulomb pre-stress and spatial heterogeneity can and should be calculated by considering all known past earthquakes, interseismic loading from underlying shear zones and the geometry of the active faults, while knowledge of fluid pressure changes may not be required to explain known sequences of earthquakes. Importantly, it does appear that earthquakes tend to occur on positively stressed faults, both where the majority of the fault surface is positively stressed, or where high stress patches exist on faults with negative mean Coulomb stress, once interseismic loading and local stress concentration on non-planar faults is taken into account. Our findings agree with the conventional Coulomb triggering hypothesis if cumulative CST is studied, and we introduce two new measures to assess this hypothesis; the maximum CST on a single fault patch and the proportion of the fault that is positive. In our study sample of 34 earthquakes over a period of 667 years, earthquakes tend to nucleate on sections of the active faults where Coulomb stress is positive, propagating both across faults that are positively stressed, and in a few examples, from positions where highly-stressed patches are surrounded by regions of the fault plane that are negatively stressed. More work is needed to examine other earthquake sequences to see if our findings apply for all tectonic settings.

## Methods

**Modeling non-planar normal faults**. Reference [11] details the method used to generate strike-variable fault planes from surface fault traces, which are based on extensive fieldwork and satellite imagery[22–27,52–57] (Supplementary Data 5). Faults are modeled as a series of 1 km rectangular elements that make up the non-planar fault surface. All CST calculations are undertaken in Coulomb 3.4[28,58].

**Comparing planar and strike-variable CST models**. To investigate the impor-tance of including the strike-variable geometry in CST calculations, four recent earthquakes are modeled using planar and strike-variable fault geometry. The selected earthquakes are modeled with the equivalent magnitude for comparison. The difference is calculated by subtracting the planar CST values from the strike-variable CST values for each fault element, therefore a positive difference means that the strike-variable model has greater magnitude CST.

**Modeling historical coseismic and interseismic CST**. Thirty four historical earthquakes are modeled on 41 faults, following ref. [10], with some additional earthquakes in the northern Apennines included[20]. Historical earthquakes are modeled with a concentric slip distribution as there is a lack of available information[11,37]. The sensitivity of the coseismic CST to the slip distribution used (Supplementary Data 3) has been tested and it is shown that the pattern of positive and negative stress is relatively consistent across all models, the magnitude of the Coulomb stress transferred differs between the models. Interseismic CST is mod-eled using shear zones from 15–24 km underlying the brittle portions of faults[10,21], the annual rate of slip on these shear zones is determined by the Holocene throws measured at the surface[22–27,54–57,59] (Supplementary Data 5).

**Calculating cumulative CST**. It is assumed that the stress on all faults in 1349 A.D. is zero[10,60,61] in the absence of any information about pre-stress prior to this date. Coseismic CST and annual interseismic CST is summed for each 1 km element of fault plane at each time point prior to an historical earthquake occurring. When an earthquake occurs, it is assumed that all the accumulated stress is released and the stress on the fault that slips reduces to zero.

## Data availability
The source data underlying Fig. 4 are provided in Supplementary Data 4.

## Code availability
The code for generating strike-variable fault geometry is available at https://github.com/ZoeMildon/3D-faults or by contacting the corresponding author.

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

## Acknowledgements

This study was funded by the UK Natural Environmental Research Council (NERC) Studentship NE/L501700/1, NERC Urgency Grant NE/P01660X/1 (EEFIT Reconnaissance Mission to the Amatrice, Italy, 24/09/2016 Earthquake), and NERC Standard Grant NE/I024127/1. The method to model strike-variable faults was developed during a

Japan Society for the Promotion of Science (JSPS) Short Term Fellowship PE15776 undertaken by the lead author with ST. Work was aided by a GBSF (Great Britain Saskawa Foundation) grant 4602 to JFW. Figures were generated using Generic Mapping Tools using 10m DEM available. We thank those who assisted with fieldwork (listed in alphabetical order), Andrew Watson, David Arnold, Francesco Iezzi, Laura Gregory, Luke Wedmore, Ken McCaffrey, and Peter Mildon, even if their opinions differ from our own.

## Author contributions

Z.M. undertook all the Coulomb stress modelling, Z.M. and S.T. developed the approach to modelling strike-variable faults, G.R., J.F.W. and Z.M. collected field data on the locations of active faults, Z.M., J.F.W. and G.R. interpreted the results of the stress modelling with respect to the historical record. All authors contributed to finalizing and editing the manuscript.

## Additional information

**Competing interests:** The authors declare no competing interests.

