## [Peer Review File · Nature Communications]

Reviewers' comments:

Reviewer #1 (Remarks to the Author):

June 20, 2018

Review of 'Coulomb pre-stress and fault bends: ignored yet vital factors for earthquake triggering' submitted to Nature Communications by Mildon et al.

I (Ruth Harris) can be identified to the authors.

This manuscript describes how it's much better to conduct long-term stressing rate models of fault systems than to just do single earthquake Coulomb stress change calculations. I agree in principle. A few things that I wasn't sure about include how non-elastic effects might affect the conclusions, including long-term viscoelastic response to centuries of earthquakes in the fault system, and additionally, shorter term plastic (or other) responses in the vicinities of the fault bends. These aspects may perhaps be sufficiently represented by elastic models, but sometimes a non-elastic solution is needed. I would like to see a bit more discussion about these topics. I recommend minor revisions.

Introductory Paragraph (lines 16-32):

Please revisit this paragraph. It isn't written as well as the later text and some of the words in the paragraph need to be revised. For example, please check the sentences on lines 18-20. Are they saying what the authors intended?

The largest negative or positive Coulomb stress changes occur on or close to the fault(s) that slipped during a large earthquake.

Also, the idea of stress concentrations due to non-planar fault geometry has been around for awhile (i.e., since at least the 1980's, if not earlier), so I think that the last sentence in the abstract should be softened to something like 'Stress concentrations on strike-variable faults are an important factor for earthquake triggering'.

Figures:

Figure 1 is really nice.

Figure 2 is very good. It is a complicated figure that works to display a lot of information. One thing that I was unsure about was if the fault bends lead to stress concentration, or not. I'm thinking that they should lead to stress concentrations. It will however depend a lot on how the faults are loaded in the model.

Figure 3 general detail question:

Maybe I misread this – in figure 3, is a negative CST indicative of future fault failure? I'm used to positive CST being indicative of future fault failure, and negative CST indicating a stress shadow. The caption words say this how I'm thinking of it, but I'm confused about the blue colors (negative CST)

in the figure for the 'ruptured in 2016' fault segments. Or is this figure the calculation for right after the 2016 fault segments earthquakes?

Movies – I like the movies. They do a good job of driving home the point about the effects of long-term stressing versus single-event CTS calculations. The CTS-alone movie appears to show disjointed ruptures popping off in mysterious locations. The long-term stressing movie reminds the viewer that tectonic loading is what really counts and that CTS is just an additional nudge.

Reviewer #2 (Remarks to the Author):

This paper compares the contributions of inter seismic loading, taking into account fault geometry with a pure Coulomb stress transfer approach to understanding future fault ruptures that produce earthquakes with $M_w \geq 5.5$. This process is undoubtedly important - but there remain some questions regarding the analysis in the paper.

1. The paper cites the coincidence of more 31 events out of 41 as being evidence of this process increasing the predictability of the location of future earthquakes as compared to the purely coulomb stress transfer model. The magnitude of the stress transfers when earthquakes occur is summarised in Figure 4. Whilst I cannot give a fully quantitative analysis of this figure, it is important to note that fig 4a shows an average of zero for the mean coseismic stress change whilst figure 4c the mean is increasing to modern times. It is therefore unsurprising that most events initiate in positively stressed regions - as most regions are positively stressed. I think the statistics need presenting in a more transparent and convincing manner here.

2. Building on this, I think that the authors need to elaborate on how well converged the obscured sample of the mean cumulative co and interseismics stress analysis is. This mean is increasing from left to right in figure 4c and the maximum is increasing in figure 4d. These suggest that the stress distribution has not yet reached a time averaged steady state.

This is important for a few reasons including a) if all the cells in figure 1c became positively stressed there would be a 100% correlation between positive stresses and the occurrence of earthquakes but no gain in predictability - this is clearly not the desired endpoint. b) In a converged distribution, it would seem appropriate to reference to the long term average rather than a zero stress datum - note that several of the earthquakes (blue dots) occurring the the lower quartiles of the means stresses (black dots) in Figure 4c and c) given the importance the authors cite in cumulative stress explaining the modern trends - this assessment of the unaccounted for stress contribution should be reflected in the strength of the predictions for modern seismicity.

The way the paper is written suggests that the data should in principle be able to predict the future events. At the same time it cites the need for historic loading which we know has not reached a long term steady state. I think this is a bit contradictory.

3. Holocene throws have been used to determine annual slip rates for use in calculating inter seismic CST. When you compare the interseismic maps with the slip rates are there any inactive faults lying within positive areas or is there complete self-consistency?

In summary, I think that more work is needed on the statistical analysis and presentation to ensure the detail in the results are not artefacts of a temporally increasing mean stress across the model. I am not saying that the premise of the fault geometry and long term loading is important - but that I find it hard to see through the data given the non-stationary mean stress to substantiate the detailed and nuanced narrative about the behaviour of the faults.

Reviewers' comments:

We thank the reviewers for their detailed comments and we have revised and improved on the manuscript. Detailed responses to the reviewers are below. We have also increased some of the analysis of the results because of the additional word limit in Nature Communications (compared to Nature Geoscience where this manuscript was first submitted).

Reviewer #1 (Remarks to the Author):

June 20, 2018

Review of 'Coulomb pre-stress and fault bends: ignored yet vital factors for earthquake triggering' submitted to Nature Communications by Mildon et al.

I (Ruth Harris) can be identified to the authors.

This manuscript describes how it's much better to conduct long-term stressing rate models of fault systems than to just do single earthquake Coulomb stress change calculations. I agree in principle. A few things that I wasn't sure about include how non-elastic effects might affect the conclusions, including long-term viscoelastic response to centuries of earthquakes in the fault system, and additionally, shorter term plastic (or other) responses in the vicinities of the fault bends. These aspects may perhaps be sufficiently represented by elastic models, but sometimes a non-elastic solution is needed. I would like to see a bit more discussion about these topics.

We have added a discussion section to the text from line 286 – 311 to address the comment above. We include discussion of a recent paper about post-seismic stress changes in the Apennines (published since the first submission of this manuscript) and the implications of the study on the interpretation of our results. We also make it clearer on line 110 that we are not considering dynamic nor post-seismic stresses and why we are not considering these factors, and on lines 119-123 we highlight the difference in magnitude between pre-stress and coseismic/post-seismic/dynamic stress triggering.

I recommend minor revisions.

Introductory Paragraph (lines 16-32):

Please revisit this paragraph. It isn't written as well as the later text and some of the words in the paragraph need to be revised. For example, please check the sentences on lines 18-20. Are they saying what the authors intended?

Thank you for highlighting this. We have considerably rewritten the abstract on lines 17 – 28 to make our key points clearer.

The largest negative or positive Coulomb stress changes occur on or close to the fault(s) that slipped during a large earthquake.

Yes we agree with this statement and have added text on line 31 to make this point clearer.

Also, the idea of stress concentrations due to non-planar fault geometry has been around for awhile (i.e., since at least the 1980's, if not earlier), so I think that the last sentence in the abstract should be softened to something like 'Stress concentrations on strike-variable faults are an important factor for earthquake triggering'.

We have removed the reference to strike-variable faults in this sentence to focus on the importance of Coulomb pre-stress and we acknowledge the prior work on this topic with additional references in line 43.

Figures:

Figure 1 is really nice.

We are glad the reviewer appreciates Figure 1.

Figure 2 is very good. It is a complicated figure that works to display a lot of information. One thing that I was unsure about was if the fault bends lead to stress concentration, or not. I'm thinking that they should lead to stress concentrations. It will however depend a lot on how the faults are loaded in the model.

There are examples of stress concentrations formed by fault bends in Figure 2, the bottom row of panels shows the difference between the planar and non-planar models. By stress concentrations, we mean that the stress is elevated or decreased by the presence of the bend. We agree that the coseismic CST is dependent on how the faults are loaded, both in terms of the interseismic loading and the coseismic slip distribution used. We have included a sensitivity analysis of the coseismic CST for a variety of slip distributions in ES6. This shows that the regions of positive and negative stress remain consistent across all models – including stress concentrations at bends – the magnitude of stress is more variable. Therefore our general assertions remain. We have added text to lines 360 – 363 in the Methods section and lines 151-153 to make this point clearer.

Figure 3 general detail question:

Maybe I misread this – in figure 3, is a negative CST indicative of future fault failure? I'm used to positive CST being indicative of future fault failure, and negative CST indicating a stress shadow. The caption words say this how I'm thinking of it, but I'm confused about the blue colors (negative CST) in the figure for the 'ruptured in 2016' fault segments. Or is this figure the calculation for right after the 2016 fault segments earthquakes?

Negative CST is indicative of a stress shadow, following the typical interpretation of negative CST. The faults that ruptured in the 2016 in Figure 3b have both positive CST at depth (>~8km) and negative CST at shallow (<~8km) depths- this is one of the interesting observations of this earthquake sequence that the authors have published on before (this reference has been added to the text). This stress pattern is also robust for a range of slip distributions (see example in ES6) We would suggest that the earthquake was able to nucleate at depth (i.e. in the positive CST region) and the rupture was able to spread across the fault plane, rupturing negatively stressed regions. We have added text from line 147 – 163 to make this point clearer.

Movies – I like the movies. They do a good job of driving home the point about the effects of long-term stressing versus single-event CTS calculations. The CTS-alone movie appears to show disjointed ruptures popping off in mysterious locations. The long-term stressing movie reminds the viewer that tectonic loading is what really counts and that CTS is just an additional nudge.

Thank you for your comment and appreciation of the videos.

Reviewer #2 (Remarks to the Author):

This paper compares the contributions of inter seismic loading, taking into account fault geometry with a pure Coulomb stress transfer approach to understanding future fault ruptures that produce earthquakes with Mw_5.5. This process is undoubtedly important - but there remain some questions regarding the analysis in the paper.

1. The paper cites the coincidence of more 31 events out of 41 as being evidence of this process increasing the predictability of the location of future earthquakes as compared to the

purely coulomb stress transfer model. The magnitude of the stress transfers when earthquakes occur is summarised in Figure 4. Whilst I cannot give a fully quantitative analysis of this figure, it is important to note that fig 4a shows an average of zero for the mean coseismic stress change whilst figure 4c the mean is increasing to modern times. It is therefore unsurprising that most events initiate in positively stressed regions - as most regions are positively stressed. I think the statistics need presenting in a more transparent and convincing manner here.

Firstly, To ensure that our analysis is transparent we provide an Excel spreadsheet of the data included in Figure 4 for inclusion in Supplementary Material (ES5). Additional analysis and numbers given in the text and in the response to the reviewers come from this spreadsheet.

Secondly, we agree that the mean stress appears to increase over time in Figure 4c. However, we have added new text to lines 225 - 260 that introduce 3 points that we think show that we can assess where earthquakes occur within the heterogenous field of cumulative CST produced by fault and shear zone interaction, even if full steady state has not been achieved since 1703 A.D.. The 3 points made show that interaction between faults and shear zones is underway by 1703 A.D., and that the cumulative CST values we study are not atypical of an interacting extensional system. For the sake of brevity, we respectfully ask the reviewer to read the 3 points in the new text rather than repeating them in this rebuttal. So, to re-iterate, we think can do what we want to do, and assess where earthquakes occur within the heterogenous field of cumulative CST produced by fault and shear zone interaction.

Thirdly, we can re-assure the reviewer that it is not simply the case that “most events initiate in positively stressed regions – as most regions are positively stressed”.. Below we show two plots: the first plot is the percentage of all faults in the area studied that have positive mean cumulative pre-stress. The percentage is fairly constant between 70 – 76% especially considering values beyond 1703 A.D. (to reduce bias to initial conditions of the model). This means that it is not the case that more faults are becoming positively stressed through the time period we examine. If anything, the percentage is slightly decreasing as across strike interaction produces more stress shadows with reduced CST values. This is point 2 that we have included in the text on lines 237 – 240. This plot has been added to Figure 4 and additional text included in the caption on lines 434 - 438.

The second plot is for faults with positive mean coseismic CST. The percentage is more variable than plot 1; this is because the percentage of faults that are positively stressed from coseismic CST alone is more dependent on the local fault geometry and the number of faults along (positive coseismic CST) and across (negative coseismic CST) strike of the fault that ruptures. The key point for this 2nd plot is that values are higher than for the 1st plot; in other words more faults are positively stressed when only the coseismic CST is considered, the opposite of what the reviewer was concerned about. Again, this also implies that it is not the case that more faults are becoming positively stressed through the time period we examine. This plot has been added to Figure 4 and additional text included in the caption on lines 422 - 426.

2. Building on this, I think that the authors need to elaborate on how well converged the obscured sample of the mean cumulative co and interseismics stress analysis is. This mean is increasing from left to right in figure 4c and the maximum is increasing in figure 4d. These suggest that the stress distribution has not yet reached a time averaged steady state.

Thank you for commenting on this, we have changed our analysis and now only analyse earthquakes beyond 1703 A.D., this allows for burn-in time for our models and reduces the bias towards the initial conditions. We choose 1703 A.D. for 3 reasons which we have outlined in the text in lines 225 - 260. The new text states that we believe that full convergence has not occurred (we refer to this as steady-state), but that our results do not rely on full steady-state as the effects of interaction between shear zones and faults are underway and clear in the data after 1703 A.D. (our new points 1-2), and our values of cumulative CST are not atypical of an interacting system because there is no obvious relationship between them and time (our new point 3).

We add further clarification by stating that it is unlikely the system has reached steady state, because not all faults have had earthquakes occur on them, even with the longest complete historical record we know of for the Earth. But we make this caveat in the text on lines 227 – 232.

Also, the statement about the “maximum is increasing in figure 4d” is not correct because there was no time scale included in Figure 4d. To help clarify this to the reader, we have now colour coded the dots in Figure 4b and 4d according to date; this shows that there is no systematic change through time on the magnitude of the maximum cumulative CST. We have added this to the figure caption on lines 444 - 446 and it is addressed in our new point 3 on lines 240 - 244.

This is important for a few reasons including a) if all the cells in figure 1c became positively stressed there would be a 100% correlation between positive stresses and the occurrence of earthquakes but no gain in predictability - this is clearly not the desired endpoint.

We agree with this statement and our new points 1-3 in the text address this point. Also, note that what the reviewer points out would never happen because earthquakes will always generate stress shadows (regions of negative coseismic CST) and therefore there will always be cells that are negative. We have analysed the proportion of faults that are positively and negatively stressed in the time period 1703 – 2016 and we find that the proportion of positively stressed faults is approximately constant, ranging from 70 - 76 %. We have added text on lines 237 – 240 and have added new plots to Figure 4.

b) In a converged distribution, it would seem appropriate to reference to the long term average rather than a zero stress datum - note that several of the earthquakes (blue dots) occurring the lower quartiles of the means stresses (black dots) in Figure 4c and c) given the importance the authors cite in cumulative stress explaining the modern trends - this assessment of the unaccounted for stress contribution should be reflected in the strength of the predictions for modern seismicity.

The zero stress datum was included to make it clearer the distinction between positive and negative stress values. We have kept the line for this purpose. We have explain above in our new points 1-3 that the system is unlikely to have reached steady-state, but this is not necessary for our conclusions.

The way the paper is written suggests that the data should in principle be able to predict the future events. At the same time it cites the need for historic loading which we know has not reached a long term steady state. I think this is a bit contradictory.

We agree that the area we study is unlikely to have reached steady state, and this is now stated this in the text. We have re-phrased it to say that our aim is to assess where earthquakes do occur within the heterogeneous field of cumulative CST produced by fault and shear zone interaction. Our 3 new points in the text suggest we can achieve this, even without steady state being achieved (see above and the new text). We certainly never wanted to claim that we can predict the locations of earthquakes in this paper, and we think our revised text reflects this, and instead suggests that earthquake appear to occur on both fully positively stressed faults and those with mean negative stress but including positively stressed patches. If anything, this makes it even harder to predict earthquakes, not easier, and we hope our new text reflects this. However our results show that when the pre-stress is considered, 76% of faults have positive pre-stress greater than 0.1 bars which adds validity to the argument that earthquake rupture is related to the CST. Our results could be used to identify faults as less likely to rupture is the pre-stress is small or negative.

3. Holocene throws have been used to determine annual slip rates for use in calculating inter seismic CST. When you compare the interseismic maps with the slip rates are there any inactive faults lying within positive areas or is there complete self-consistency?

We find this statement unclear. We have interpreted it to be asking if any of the faults we have included are inactive and positively stressed. We define active faults in the region as having Holocene (<15,000 years old) offsets that can be observed in the field. We are able to map faults that have small offsets in the field (~1.5m, which would correspond to 0.1mm/yr throw rate). All faults that we define to have offsets, and are long enough to penetrate the seismogenic layer (15km depth), and so have an associated shear zone. All faults that are mapped as active based on field data are included in the models. Therefore, there is complete self-consistency within our models. So the short answer is “No”, faults that

are not active do not have positive interseismic loading because if the fault is inactive so is the underlying shear zone in our models, i.e. we do not include it in our model. Note we do include some faults with shorter lengths.

In summary, I think that more work is needed on the statistical analysis and presentation to ensure the detail in the results are not artefacts of a temporally increasing mean stress across the model. I am not saying that the premise of the fault geometry and long term loading is important - but that I find it hard to see through the data given the non-stationary mean stress to substantiate the detailed and nuanced narrative about the behaviour of the faults.

We hope that there is a spelling mistake above, we hope that the reviewer meant to say that “not saying that the premise of the fault geometry and long term loading is unimportant”. Through the points answered above we have made our statistical reasoning clearer and included more numerical values of the reasoning where appropriate in the text.

Reviewers' comments:

Reviewer #1 (Remarks to the Author):

February 28, 2019

Review of the manuscript revised for Nature Communications 'Coulomb pre-stress and fault bends: ignored yet vital factors for earthquake triggering and seismic hazard' by Mildon, Roberts, Faure Walker, and Toda.

I (Ruth Harris) can be revealed to the authors.

The revised manuscript is in good shape, I just have a few comments, many of which are indicated in the annotated text.

Specific comments:

Maybe I missed it, but does the mathematical definition of Coulomb stress transfer (CST) appear anywhere in the text? If no, please add it. (e.g., use handy simple equation from Harris and Simpson, Nature, 1992 or Harris, JGR, 1998, etc.)

Line 22. I think that the CST can be high (more than 2 bars) where faults are close together. I'm not sure where the 2 bars number comes from. Is it an average over some distance? On the other hand, I completely agree that the important factor in earthquake occurrence in tectonically active regions is the fast tectonic loading rates on the faults. It is primarily in slowly deforming continental regions that the CST appears to have the potential to be a dominant factor.

Line 38. 'and assumed to be zero'. I don't think that this is correct, so please omit this phrase, and the related text in the next lines.

I've indicated how to do so in the annotated text.

Also, this relates to Line 367 in the text.

Line 337. I'm a little confused by the phrase 'negatively stressed fault plane'. Is the long-term tectonic loading putting this fault plane into a stress shadow?

Figure 4. I'm thinking that subfigures c and f could have similar y-axis scales.

Overall question – the potential roles of pore-pressure changes aren't mentioned in the text. Is this o.k.?

We have addressed the reviewers comments individually below. The new text in the manuscript file is highlighted in blue, to distinguish it from earlier revisions in red. In addition to the changes detailed below we have made minor changes to wording elsewhere in the text which are also highlighted in blue text.

Review of the manuscript revised for Nature Communications 'Coulomb pre-stress and fault bends: ignored yet vital factors for earthquake triggering and seismic hazard' by Mildon, Roberts, Faure Walker, and Toda.

I (Ruth Harris) can be revealed to the authors.

The revised manuscript is in good shape, I just have a few comments, many of which are indicated in the annotated text.

Specific comments:

Maybe I missed it, but does the mathematical definition of Coulomb stress transfer (CST) appear anywhere in the text? If no, please add it. (e.g., use handy simple equation from Harris and Simpson, Nature, 1992 or Harris, JGR, 1998, etc.)

No we had not included the equation defining Coulomb stress transfer, thank you for the suggestion. We have included it and the references suggested on lines 58 - 64.

Line 22. I think that the CST can be high (more than 2 bars) where faults are close together. I'm not sure where the 2 bars number comes from. Is it an average over some distance? On the other hand, I completely agree that the important factor in earthquake occurrence in tectonically active regions is the fast tectonic loading rates on the faults. It is primarily in slowly deforming continental regions that the CST appears to have the potential to be a dominant factor.

We agree that the CST can be greater than ± 2 bars. This value is chosen because 98% of mean coseismic CST values for all faults analysed in the Apennines are within this range for the data set studied in this paper (see histogram below).

We have added this histogram in ES6 (data table of all CST values calculated within the manuscript) to demonstrate this point and referred to this on line 80.

Line 38. 'and assumed to be zero'. I don't think that this is correct, so please omit this phrase, and the related text in the next lines.

I've indicated how to do so in the annotated text.

Also, this relates to Line 367 in the text.

We partially disagree with the reviewer on this point. Most studies analyse a single earthquake and the CST onto the surrounding faults – and from this postulate which fault will be the next one to experience an earthquake. We are arguing strongly against this approach, and hence it is important to keep this point in the manuscript. We have rephrased the relevant sentence to make it clear that this approach is problematic for analysing seismic hazard in particular, on lines 38 - 39.

However, we also acknowledge that not all authors do this, some studies frame the amount of the CST related to the tectonic stressing rate or fault slip rate and calculate the temporal magnitude of advance or delay of an earthquake. We have added this point and references to support this on lines 39 - 41.

Line 337. I'm a little confused by the phrase 'negatively stressed fault plane'. Is the long-term tectonic loading putting this fault plane into a stress shadow?

That is not what we meant by this line. We have reworded it on Line 354 to clarify our point, to read "regions of the fault plane that are negatively stressed" – i.e. a single fault surface can have both positively and negatively stressed regions (as shown by our results).

Figure 4. I'm thinking that subfigures c and f could have similar y-axis scales.

We have changed the scales so both figures go from -10 to 130 bars. We have chosen to use this as a linear scale because we cannot represent the negative values on a logarithmic scale. We have added a sentence in the caption to this figure to explain on lines 448 – 449.

Overall question – the potential roles of pore-pressure changes aren't mentioned in the text. Is this o.k.?

We now discuss the possible role of pore fluid pressure in the manuscript. We have neglected fluid pressure changes because (1) we lack direct measurements of this factor at seismogenic depths, and (2) we are interested to see if we can explain our observations without adding *ad hoc* fluid pressure changes. We have neglected fluid pressure changes in this study and yet manage to explain our observations of ruptures circumventing nearest neighbour faults by using the constraints we place on pre-stress. This suggests that fluid pressure changes are not required to produce the effects we have described, although we have not ruled out some fluid pressure influences, and this requires more study. We have added text to explain this on lines 64 - 69, 320-326 and 341 - 342.